# A GOODNESS OF FIT MEASURE FOR GENERATIVE NETWORKS

## ABSTRACT

We define a goodness of fit measure for generative networks which captures how well the network can generate the training data, which is necessary to learn the true data distribution. We demonstrate how our measure can be leveraged to understand mode collapse in generative adversarial networks and provide practitioners with a novel way to perform model comparison and early stopping without having to access another trained model as with Frechet Inception Distance or Inception Score. This measure shows that several successful, popular generative models, such as DCGAN and WGAN, fall very short of learning the data distribution. We identify this issue in generative models and empirically show that overparameterization via subsampling data and using a mixture of models improves performance in terms of goodness of fit.

## 1 INTRODUCTION AND RELATED WORK

Generative adversarial networks (Goodfellow et al., 2014) are a specific type of generative model that has shown impressive performance lately. The main idea is that there are two networks that compete against each other: a generator network that generates images and a discriminator network that tries to distinguish between real and fake images. These models are useful because they can generate very realistic images that are not in the training set. Throughout the rest of the paper, we will use GANs as a specific class of models to study, however, the goodness of fit measure discussed in Section 3.2 and its applications can be extended to other generative networks such as Variational Autoencoders.

Some GANs that appear to be successful in practice cannot actually reproduce the training set, as we will see. Other generative models, such as Generative Latent Optimization (GLO) (Bojanowski et al., 2017) and Implicit Maximum Likelihood Estimation (IMLE) (Li and Malik, 2018) attempt to memorize the training data as part of the learning algorithm. These methods are not as successful as GANs in producing realistic images. We believe that the reason for this difference in performance is due to a lack of overparameterization in GANs, GLO, and IMLE.

Our solution starts with measuring how well a generative model can generate the training data. We explain in Section 3.2 that our goodness of fit measure $F(G)$ is zero if we are able to perfectly generate the training data. If we cannot generate the training data, then $F(G)$ represents how far away we are from generating our training set in an average of total least square sense. We use this goodness of fit measure to evaluate different models and training settings as well as study the evolution of the approximation error through training.

Ideally, we would like to overparameterize GANs in order to increase their capacity and reduce $F(G)$. Recently, it was shown that overparameterization in classifiers and autoencoders leads to better performance (Radhakrishnan et al., 2019; Belkin et al., 2018). Another reason to overparameterize is that we observed, while calculating $F$, that our models actually use the the full potential of the latent distribution on $z$ to generate different images. In other words, suppose that we train a GAN with $z \sim \mathcal{N}(0, I)$, then we observe that the optimal $z$'s corresponding to the closest generated images from the training set are also distributed as $\mathcal{N}(0, I)$. That is, once trained, the latent distribution fixed a priori becomes the optimal one minimizing the approximation error.

Despite the above findings, increasing the complexity of the generator becomes very difficult due to the training algorithms for GANs requiring careful hyper-parameter settings for convergence (Rege

and Monteleoni, 2019). As such, we explore two alternatives that do not impact the training stability. First, increasing the dimension of the latent space which is currently set to 100 across models and dataset. We demonstrate that this solution indeed allows to reduce the approximation error. Second, we consider a mixture of GAN setting. That is, we train $K$ different GANs on subsets of the data of size approximately $\frac{N}{K}$ for total data size $N$. Note that training a mixture of GANs has been done in practice Hoang et al. (2017), we thus will quantify the approximation error reduction that can be obtained with this solution. Hence, we see that if our original GAN has $P$ parameters, we now have $KP$ parameters total. Also, each one is trained on $N/K$ data. Hence, we see that our "effective" parameters for this mixture of GANs is $\frac{P'}{N'} = K^2 \frac{P}{N}$. Hence we can get an effective overparameterization of 100 fold if we divide our data into $K = 10$ subsets. We first demonstrate how a single GAN trained on a smaller dataset has a smaller approximation error. In particular we also find that how the dataset is subsampled, random or from a clustering based partitioning, matters for performances. We then build on those finding to train the mixture of GANs on a K-means based partitioned dataset and demonstrate important reduction in the approximation error.

We summarize our contributions that apply to arbitrary generative models s.a. GANs, VAE as follows:

- We provide a novel goodness of fit measure for generative networks and how it defines necessary conditions for generative networks to be optimal (Sec. 3.1). We also relate the metric to mode collapse and provide implementation details on computing it efficiently (Sec. 3.2). Finally we demonstrate how our metric compares to standard GAN metric s.a. the Frechet score (Sec. 3.3).

- We demonstrate how our goodness of fit metric allows to gain novel insights into GANs. We show that DCGAN and WGAN do not memorize and have very different behavior w.r.t. overfitting Sec. 4.1. In particular DCGAN is able to match with WGAN performances if early stopping is performed. We then show the impact architecture and residual connection (Sec. 4.2). Finally, we study the latent space distribution and in particular demonstrate how the optimal latent space distribution which minimizes the approximation error of a trained GAN matches the distribution used for training, highlighting how current approximation errors are due to underparametrized GANs (Sec. 4.3).

- We provide two solutions to reduce the approximation error without altering training stability Sec. 5. First, we propose to increasing the latent space dimension in Sec. 5.1. Then we study how dataset subsampling also helps reducing the approximation error Sec. 5.2 which motives the use of a mixture of GANs Sec. 5.3.

## 2 BACKGROUND

In this section we briefly overview Generative Adversarial Networks (GANs) and generative latent optimization (GLO) (Bojanowski et al., 2017), which is another generative model. Finally describe how one can optimize the generator network latent space to obtain a desired generated sample. We remind that all our development applies to arbitrary black/white box generative networks even though we only focus on GANs in this paper.

**Generative Adversarial Networks.** Generative adversarial networks (GANs) are generative neural networks that use an adversarial loss; the adversarial loss is typically another neural network. In other words, a GAN consists of two neural networks that compete against each other. The generator network $G : \mathbb{R}^\ell \to \mathbb{R}^p$ generates $p$-dimensional images from an $\ell$-dimensional latent space. The discriminator $D : \mathbb{R}^p \to (0, 1)$ is a classifier which is trained to distinguish between the training set and generated images. The training loss for a batch size of $N_B$ for the discriminator is given by

$$\mathcal{L}_D = -\frac{1}{N_B} \sum_{i=1}^{N_B} \log(D(\boldsymbol{x}_i)) - \frac{1}{N_B} \sum_{j=1}^{N_B} \log(1 - D(G(\boldsymbol{z}_j))),$$

where $\boldsymbol{x}_i$ and $G(\boldsymbol{z}_i)$ is a real image and a generated image for each $i \in \{1, 2, \ldots, N_B\}$, respectively. The generator loss is given by

$$\mathcal{L}_D = \frac{1}{N_B} \sum_{j=1}^{N_B} \log(1 - D(G(\boldsymbol{z}_j))).$$

Notice that the loss for the generator does not explicitly use the training data. Instead, the training data is used indirectly through the training of the discriminator.

In this paper, we discuss two popular GANs: DCGAN (Radford et al., 2015) which uses the loss above and WGAN (Arjovsky et al., 2017) which uses a slightly different learning algorithm.

**Generative latent optimization.** In contrast to GANs, GLO is a generative network that does not use an adversarial loss. Instead, GLO attempts to memorize the training data by using this loss:

$$\mathcal{L}_G = \frac{1}{N_N} \sum_{i=1}^{N_B} \min_{\boldsymbol{z} \in \mathbb{R}^\ell} L(G(\boldsymbol{z}), \boldsymbol{x}_i)$$

where $L$ is a loss function. In the original paper, the authors use different loss functions to demonstrate how the model differs.

**Latent space optimization.** The generator is a mapping from a latent space $\mathbb{R}^\ell$ into an image space $\mathbb{R}^p$. In the GLO paper one aims at finding a specific $\boldsymbol{z} \in \mathbb{R}^\ell$ such that the generated sample $G(\boldsymbol{z})$ is close to a target output. In particular, one picks a target as being a randomly generated GAN image from some target vector $\boldsymbol{z}^*$. That is, the target is guaranteed to lie in the span of $G$. Following this, one aims at finding the $\boldsymbol{z}$ vector that led to the target $G(\boldsymbol{z}^*)$ by

$$\hat{\boldsymbol{z}} = \arg \min_{\boldsymbol{z} \in \mathbb{R}^\ell} \|G(\boldsymbol{z}) - G(\boldsymbol{z}^*)\|_2^2. \tag{1}$$

Since the above optimization problem in non-convex, there is not theoretic guarantee of finding a global minima, in general. However, empirically it was shown that the above optimization problem is solved 100% of the time in practice (Lipton and Tripathi, 2017). We now propose to leverage the above to develop our goodness of fit measure.

## 3 GOODNESS OF FIT METRIC

In this section we first motivate and define our metric (Sec. 3.1) and provide its approximation (Sec. 3.2). We demonstrate that our measure being minimized is a necessary and sufficient condition to detect mode collapse. Finally, we study the different with current GAN measures that are the Inception Score and the Frechet Inception Score (Sec. 3.3).

### 3.1 METRIC, OPTIMUM GENERATIVE NETWORK, AND MODE COLLAPSE

The generator $G$ is a continuous mapping for any type of layer used as current layers in deep learning are all continuous. As such, the following defines the image of the generator:

$$\text{Imag}(G) = \{G(F) : F \in \mathbb{R}^\ell\}, \tag{2}$$

with $\ell$ the dimension of the latent space. Then the approximation of the true data manifold denoted as $\mathcal{X}$ with $G$ can be measure by "how far" is the span of $G$. Since the two quantities to compare are sets one solution is the following standard Total Least Square metric defines as

$$d(\text{Imag}(G), \mathcal{X}) = \int \min_z \|G(z) - x\| dx \tag{3}$$

which in practice with a finite dataset becomes our proposed measure:

$$F(G, \mathcal{X}) = \frac{1}{N} \sum_{n=1}^{N} \min_z \|G(z) - x_n\|_2 = \frac{1}{N} \sum_{n=1}^{N} m(x_n; G). \tag{4}$$

As a result, $F$ is an empirical average least square distance between reference points, the observed input, and $\text{Imag}(G)$.

Turning the above argument into a probabilistic setting, we obtain the following motivation. A generative model or network is learned to approximate some target distribution. This target distribution is itself observed via some samples, the given observations. It is common to use the likelihood as a measure of fitness for such models, and it is defined as $\mathcal{L}(\mathcal{X}) = \prod_{n=1}^{N} p(x_n)$, with $p(x_n)$

some distribution density. A necessary condition to allow maximization of the likelihood is that the samples lie in the support of the distribution, that is $p(x_n) > 0 \iff x_n \in$ Support(p), as Support(p) $= \{x \in \mathcal{X} : p(x) > 0\}$. In the case of a generative network, the distribution $p(x)$ is not easily available. However its support is directly accessible as we have

**Proposition 1.** *Support(p)* $= Imag(G)$.

As a result, from a probabilistic fitting point of view, ensuring that all the samples lie in the span of the generator is a necessary condition that must be fulfilled, which otherwise would prevent maximization of the likelihood, leading to the following result.

**Theorem 1.** *The optimal generative model in term of distribution approximation must have* $F(G, \mathcal{X}) = 0$.

The above setting also allows the following direct and intuitive result that any sample $x$ in $\mathcal{X}$ such that $m(x; G) = 0$ is a sample that can be generated, potentially with very small probability, by a generative network. We conveniently employed the $L_2$ in our metric and we will use this throughout the rest of the paper. However, notice that if $\mathcal{X} \subset \text{Imag}(G)$, the choice of the distance function is not important because $d(\boldsymbol{x}, \boldsymbol{y}) = 0$ implies that $\boldsymbol{x} = \boldsymbol{y}$ for any distance metric $d$.

**Relation to mode collapse.** We now relate the value of $m(x_n; G)$ for samples from the training set to mode collapse. We also refer to the case $m(x_n; G) = 0$ as a memorized sample and in general we refer as memorization $F(\mathcal{X}, G) = 0$. The lack of memorization in current generative networks is evident when generators experience mode collapse. Mode collapse translates into the absence of approximation of the generator to some parts of the target distribution. Suppose that the data comes from a distribution $P_X$. Then, mode collapse happens if $P_X(\boldsymbol{x}) > 0$ but $\min_{\boldsymbol{z}} ||G(\boldsymbol{z}) - \boldsymbol{x}|| > 0$. In practice, we do not have $P_X$, but we do have the empirical distribution $\hat{P}_X$, so that mode collapse will occur if we do not memorize the training data. A generative model must memorize the training data in order to avoid mode collapse. We thus obtain the following result.

**Proposition 2.** *A necessary condition to avoid mode collapse for a generative network $G$ is to memorize, that is, $F(\mathcal{X}, G) = 0$.*

We now demonstrate how one can compute our metric and in particular $m(x_n; G)$ efficiently.

### 3.2 METRIC APPROXIMATION

We now turn into the actual computation of our metric. Suppose that we have a generator neural network $G : \mathbb{R}^\ell \to \mathbb{R}^p$ that maps a latent space to a data space, such as an image space. We would like to measure how much of the data set is memorized and if not, how far off is the approximation in that region of the space.

We calculate $m(\boldsymbol{x}_n, G)$ by solving the non-convex optimization problem

$$\min_{\boldsymbol{z}} d(G(\boldsymbol{z}), \boldsymbol{x}_i)$$

for $i \in \{1, 2, \ldots, N\}$. Since this is a non-convex optimization problem, we do not have many theoretic guarantees. This optimization is computationally expensive to run on every single image. Therefore, we run it on a small subset of the data to obtain an estimate $\hat{\boldsymbol{m}}(G)$. In order to show that this estimate has low variance, we bootstrap our distance calculations to calculate the standard error of $\hat{\boldsymbol{m}}(G)$. The variance of $\hat{\boldsymbol{m}}(G)$ is shown in the figures as errorbars and is typically low enough to justify using a subset of the data. We use 100 images to calculate $\boldsymbol{m}(G)$ unless otherwise stated. Moreover, for the optimization algorithm, we pick the latent variable $\boldsymbol{z}$ and error $||G(\boldsymbol{z}) - \boldsymbol{x}||_2$ that corresponds to the smallest error instead of picking the latent variable that Adam Kingma and Ba (2014) finds. This is because we can be pushed out of local minima temporarily since we are using stochastic optimization. Nevertheless, theoretical analysis of the surface of the generator from Hand and Voroninski (2019) suggests that such optimization can be done with great chances of reaching the global minima.

### 3.3 CORRELATION WITH INCEPTION AND FRECHET SCORES

In this section highlight the key differences between our proposed measure and existing ones, in particular the Inception Score (IS) (Salimans et al., 2016) and its extension, the Frechet Inception distance (FID).

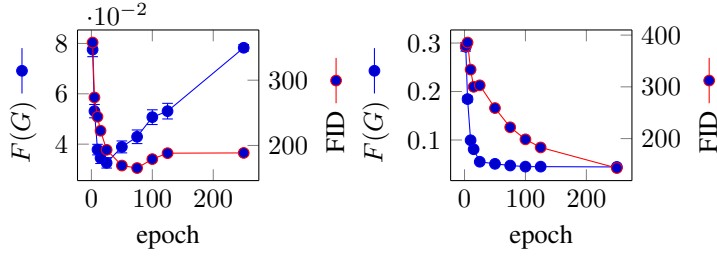

Figure 1: Depiction of our fitness measure in blue during training with yaxis on the left of the figures versus the FID measure in red with yaxis on the right of the figures (**left**) DCGAN and (**right**) WGAN.

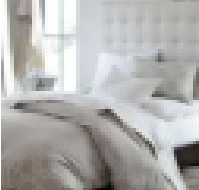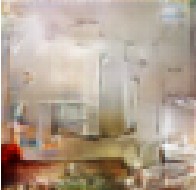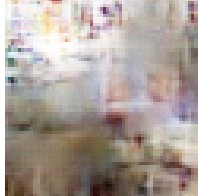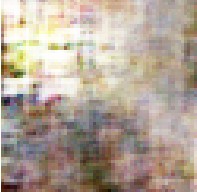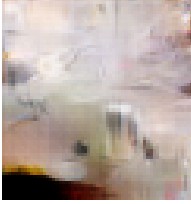

Figure 2: Depiction of the closest reconstruction to the target image (**left**) from WGAN trained as (from left to right) (i) standard, all dataset (ii) with K-means subsampling $K = 10$ (iii) with K-means subsampling $K = 100$ (iv) with 300 dimensional latent space dimension

A current limitation of metrics for GANs is their dependence on another pretrained deep neural network model. This poses limits as to what dataset GANs can be evaluated on, as well as any internal bias induced by the different models used. On the other hand our measure does not rely on any other trained model and thus can be applied as is across different datasets. We now briefly describe those metric and then empirically observe how our metric compares to FID.

The Frechet Inception distance (FID) (Heusel et al., 2017) measures how similar two data sets are. This metric is used in GAN training and evaluation because it captures, loosely speaking, how diverse the images from the generator are. However FID has two limitations: it requires a trained DN and a target dataset. This means it is sensitive to new dataset applications and to the size of the dataset. We demonstrate that our memorization metric is correlated with the FID score of a neural network in Figure 1.

We trained WGAN and DCGAN from their git repositories in order reproduce their networks. For both networks, we observe that $\hat{F}(G) > 0$ meaning that we do not observe any memorization in these GANs. It is specifically surprising that WGAN does not memorize because of the diversity of its generated images. This implies that even though a generator can produce a wide spectrum of images, it can not reproduce the training data. We provide in Fig. 2 an example of a target image and its closest reconstruction from different GAN training settings that we explore through the paper.

## 4 GENERALIZATION AND MODEL ARCHITECTURE COMPARISON

In this section we propose to demonstrate some direct applications of our metric. First we demonstrate how it can be used to probe the state of a generator during learning and measure how GAN fitness varies from training samples to validation samples. We then demonstrate how the measure can used to perform model comparison and in particular study the impact of residual connections and latent space dimension. Finally, we conclude by demonstrating how the optimal latent distribution of a trained GAN matches the distribution imposed during training.

### 4.1 VALIDATION MEASURE AND EARLY STOPPING

A key problem in GAN training is knowing when to stop. Either to save resources or prevent overfitting, being able to probe a GAN during training and infer a stopping policy is crucial. We demonstrate in this section that our goodness of fit metric can be used as a validation metric for early stopping. In fact we demonstrate how it can be used during training to provide a meaningful signal on the state of the GAN distribution approximation. We do experiments with the DCGAN

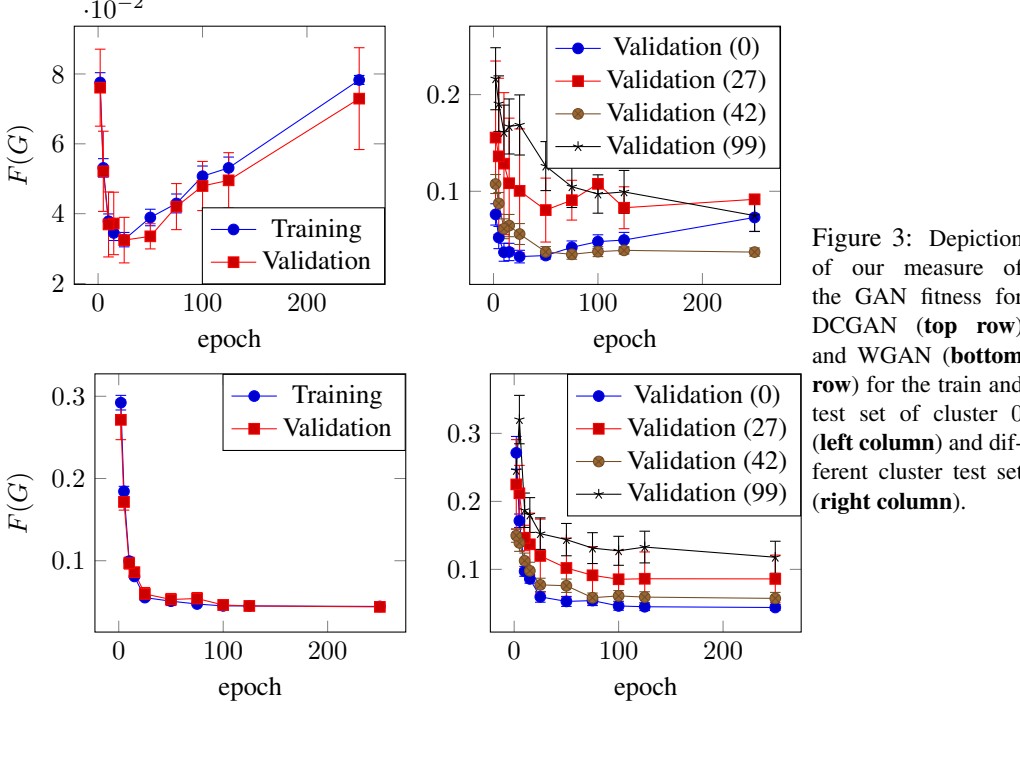

Figure 3: Depiction of our measure of the GAN fitness for DCGAN (**top row**) and WGAN (**bottom row**) for the train and test set of cluster 0 (**left column**) and different cluster test set (**right column**).

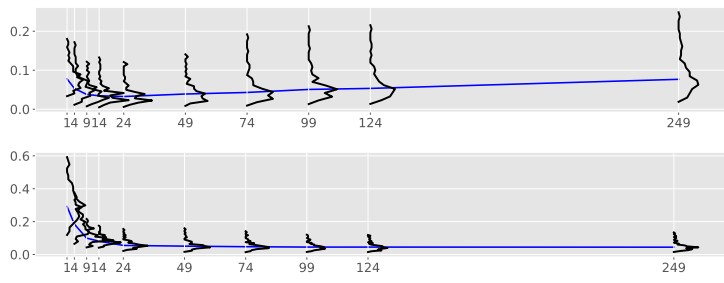

Figure 4: Evolution of the distribution of the errors $m(x_n)$ during training for DCGAN (**top**) and for WGAN (**bottom**)

model and WGAN in Fig. 3. We obtain the following conclusions. The DCGAN favors a minority of points making the overall $F(G)$ measure first decrease and then increase when this specialization occurs. In fact, the DCGAN will favor learning exactly some images and less good on average. On the other hand WGAN is much more robust to such degeneracy and thus has performances stable even if training for a long period of time. As such, in the DCGAN case our metric can be used to do early stopping and prevent decrease in overall performances, in the WGAN setting it allows to save resources by stopping when no gains appear even though the performance do not get worse with time. We also provide in Fig. 4 the evolution of the histograms of the errors $m(x_n)$ for both settings highlighting how for the DCGAN, the main reason for increase of the approximation error lies in outliers being more and more apart from Imag(G).

## 4.2 MODEL COMPARISON

In this section we propose to study the impact of the architecture onto the goodness of fit of the generator. To do so we experiment adding residual connection onto DCGAN. In Figure 7, two WGAN networks were trained on CIFAR10 (Krizhevsky et al., 2009) and the LSUN (Yu et al., 2015) bedrooms dataset.

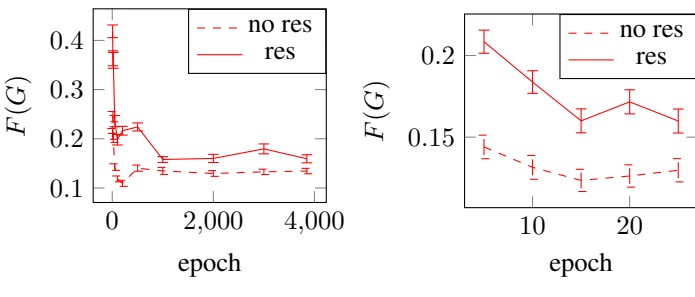

Figure 5: WGAN experiment on CIFAR10 (**left**) and LSUN (**right**) for the same network with and without residual connections

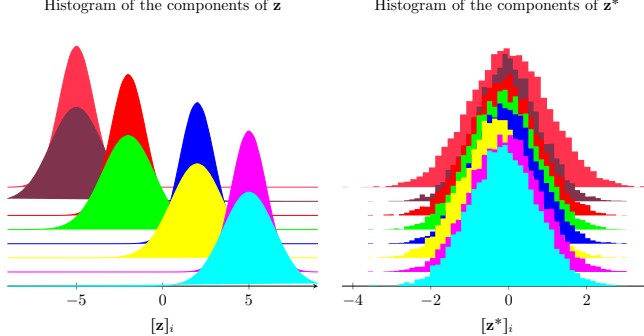

Figure 6: The initialization of $z$ and the final $z^*$. For illustrative purposed, the wide distributions on the left are shown with smaller standard deviations than in practice.

## 4.3 LATENT SPACE DISTRIBUTION ANALYSIS

Recall that we solve the optimization problem $z^* = \arg\min_{z \in \mathbb{R}^\ell} \|G(z) - x\|_2$ in order to try and generate $x$. A natural question is what is the distribution of $z^*$? In this setting we have a GAN with latent space distribution $z \sim \mathcal{N}(0, I)$. In fact, while the optimization algorithm can be initialized with normal independent and identically distributed random variables, it might be possible to have less likely $z$ that would allow better reconstruction. However we demonstrate in figure 6 that the actual distribution a GAN is trained with is also the one that will minimize the overall goodness of fit metric. That is, once trained with a specific distribution, one can not increase performance just by changing this distribution. This brings an interesting observation that the limit in the reconstruction and span of the generator comes form the model and weights themselves rather the $z$ distribution.

## 5 REDUCING A GAN'S APPROXIMATION ERROR

In this section we study the two proposed strategies to decrease the approximation error of GANs. First in Sec. 5.1 we demonstrate how the dimension of the latent space can be increased. Then in Sec. 5.2 we demonstrate how reducing the size of the dataset allows to better fit the data and then build upon this finding to experiment with a mixture of GANs in Sec. 5.3.

## 5.1 INCREASING LATENT DIMENSION

We now turn into the latent space dimension itself. It has been set to $100$ across a very large number of networks and dataset. However, recall that $\text{Imag}(G)$ depends on the latent space itself and thus incidentally on its dimension. We thus experiment with two different setting, the standard $\ell = 100$ and the expanded $\ell = 300$ and this on CIFAR10 and LSUN dataset. We report the results in Fig. 7 where one can observe how increasing the dimension naturally allows to reduce the approximation error of the generator.

## 5.2 DATASET SUBSAMPLING HELPS

We demonstrated above that current GANs can not span the training set and in general seem limited by their generative network. We propose in this section to study how dataset subsampling affects our measure. As the generator capacity is limited we reduce the complexity of the data and observe

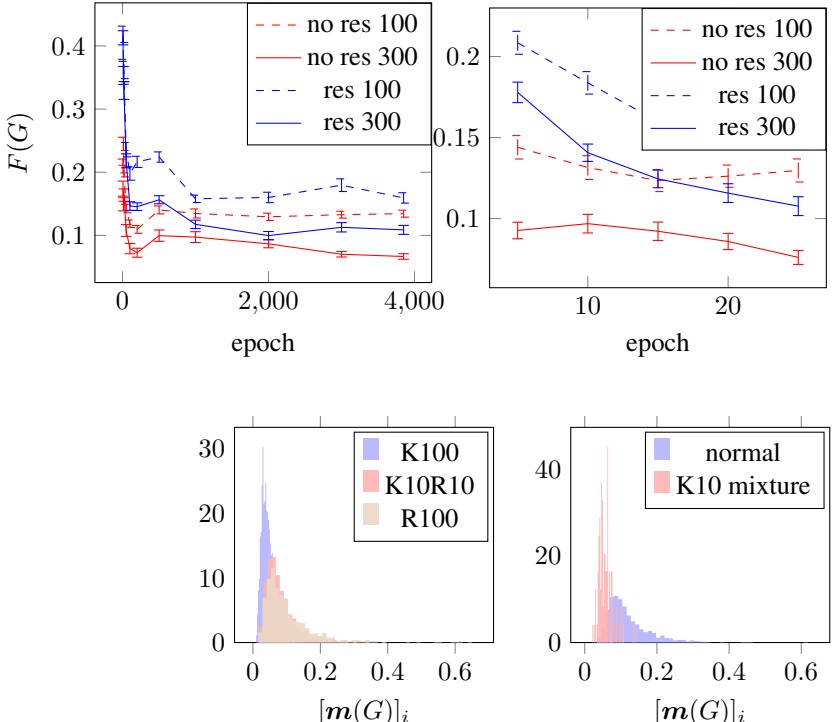

Figure 7: WGAN experiment on CIFAR10 (**left**) and LSUN (**right**) for the same network with and without residual connections.

Figure 8: (**left**): Histograms of $m(\boldsymbol{x}_i, G)$ for different subsampling methods. We notice that the K100 method is the best among the three in terms of goodness of fit. We perform random subsampling (R100), kmeans clustering (K100), and a mixture of the two (K10R10). In the K10R10 case, we first use kmeans clustering with 10 clusters and then subsample randomly by 10 in each of those clusters. (**right**) Histograms of $m(\boldsymbol{x}_i, G)$ for WGAN and our mixture of WGANs

the new measure of $F(G)$ after training. We obtain the key following conclusions. As we subsample our data and our parameters to data ratio increases, our goodness of fit measure decreases. Clustering images that are close in the input space allows for a better goodness of fit measure, even though we are in a very high dimensional setting. That is, the number of points matter but also the similarity between those points. This is a key insight that might play a role in explaining conditional GANs (Mirza and Osindero, 2014) that act as a subsampling by forcing each class to have its own set of parameters.

## 5.3 MIXTURE OF GANs

In this section we leverage the finding from the previous section and exploit a K-means based partitioning of LSUN with $K = 10$ to subsample the dataset into 10 non overlapping subsets. We then propose to train an independent WGAN on each of the subset to then form the overall generative network being a combination of the per subset WGANS. The sampling strategy thus becomes (i) randomly sample one of the subset (ii) randomly generate an image from the WGAN of the picked subset. This two step sampling can be assimilated to a mixture model such as Gaussian Mixture Model. We obtain the following approximation error when performing such training in Fig. 8.

## 6 CONCLUSION

We defined a goodness of fit measure $F(G)$ for generative networks. We used $F(G)$ to show that DCGAN and WGAN fail to memorize the training data even though they can generate compelling images. We provide means to get a better goodness of fit measure by subsampling data randomly or with K-means clustering.

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
