# OpenReview forum: "A GOODNESS OF FIT MEASURE FOR GENERATIVE NETWORKS"
_ICLR.cc/2020/Conference — Reject_

### Official Review · AnonReviewer2 · 2019-10-12
**Official Blind Review #2**

**Rating:** 1

**Review:**

The paper proposes a new goodness of fit measure for generative models, and uses it to get insight into GAN's. While this is an important topic and a novel approach, I do not think the paper delivers on what it promises.

I think this paper should be rejected. First, while it claims to be a general method for generative models it, it is limited to only  GANs and even for GANs it is limited. Second, most of the observations are nice but trivial, e.g. larger latent space leads to larger image.

Detailed remarks:
- The main point that the training set points x must have p(x)>0 under the model is naturally satisfied for almost all models except GANs such as VAEs, autoregressive model and flow models with standard implementations as the support is the whole space. This is in contrast to the claim in the paper that "its applications can be extended to other generative networks such as Variational Autoencoders.".
- Even for GANs as this measure only looks at the support and not the distribution it is not clear if this measure does more then evaluate mode collapse. While this is an important task, it falls short of the promises the authors claim.
- The authors claim that "We demonstrate that our measure being minimized is a necessary and sufficient condition to detect mode collapse." but only show that it is necessary.
- Proposition 1 is a trivial statement.
- The authors claim that " mode collapse happens if P(x) > 0 but minz ||G(z) − x|| > 0". This is a main point by the authors, but it ignores the probability and only looks at the support. It has been shown that mode collapse happens even in 2d distributions, e.g. veegan paper, where it is easy to get the support to be the whole distribution.
- The results in sec. 5 are quiet obvious, with a larger latent space you can naturally get a larger support, same as with a mixture model.


In general the method only looks at the support, ignoring the distribution over the support and is therefore very limited in evaluating generative models.


minor details:
- In eq. 3 the integration should be w.r.t dP(x) for it to be monte-carlo approximated as it is in eq. 4.
- Not 100% I understand what the authors try to say here - "we pick the latent variable z and error ||G(z) − x||2 that corresponds to the smallest error instead of picking the latent variable that Adam Kingma and Ba (2014) finds."

**Experience Assessment:**

I have published one or two papers in this area.

**Review Assessment: Checking Correctness Of Derivations And Theory:**

N/A

**Review Assessment: Checking Correctness Of Experiments:**

I assessed the sensibility of the experiments.

**Review Assessment: Thoroughness In Paper Reading:**

I read the paper at least twice and used my best judgement in assessing the paper.

---

> ### Author Response · Authors · 2019-11-13
> **Response to Reviewer #2**
>
> Thank you for the feedback on our paper. We address your concerns as follows:
>
> - For generative networks that map a latent space of “low” dimension to a space of “high” dimension, the range of the network does not cover the entire space, because two spaces are of differing dimensions. Hence, we have that p(x) = 0 can occur for training points. This is true for VAEs as well as GANs as we mentioned in the paper.
>
> - If generative models can generate high-quality images but cannot generate the training set, then the learned distribution is likely only focused on a few modes of the true data distribution. This means that the learned distribution might cover a very small part of the data distribution well, which is not desirable since we want to learn the entire data distribution. Our main contribution is the definition of a metric that enables us to measure how well a generative model memorizes the training set.
>
> - Mode collapse can be defined with respect to different distributions. With respect to the empirical data distribution, F being 0 is necessary and sufficient to avoid mode collapse. As you mention, with respect to some true data distribution that was sampled to obtain an empirical distribution, F = 0 is only necessary to avoid mode collapse.
>
> - If the latent space is much smaller than the output space, then it is not trivial to have F = 0 because the generator cannot reproduce any output image. If F = 0, then you can compare probabilities for generated images against an oracle data distribution. In many toy 2D examples, you have oracle access to the true distributions because the distributions are simulated. However, we are concerned with image datasets where we only have access to an empirical distribution. Hence, mode collapse must be compared with the empirical distribution because we do not have access to an oracle distribution.
>
> - The results with the mixture model are not obvious because they imply that using less data can lead to better performance. Specifically, this implies that DCGAN-type architectures are underparameterized.
>
> Let us know if you have any further concerns about our paper, and thank you for the helpful feedback.

---

### Official Review · AnonReviewer1 · 2019-10-21
**Official Blind Review #1**

**Rating:** 3

**Review:**

This work proposed a new goodness of fit measure for generative network evaluations, which is based on how well the network can generate the training data. The measure is zero if the network could perfectly recover the training data, and would represent how far it is from generating the training set in the average manner of the total least square sense, where the one-to-one mapping between the generated data and the training sample is constructed through latent space optimization. Using the proposed measure, the authors showed an interesting trend present in the DCGAN training and the impact of the residual connection. The authors might want to add some discussion in Section 4.2 regarding why the residual connection is detrimental for covering the support.  Increasing the model complexity through larger latent space dimension and learning mixtures is proposed as solutions to improve the measure as well.

With all the interesting results presented, I still have the concerns about the sensitivity of the proposed measure:
- It is an average over the training data or the selected sample. Above Section 4, the authors argued that "\hat{F}(G) > 0 meaning that we do not observe any memorization". This seems overly assertive. Since the measure is an average over the training data, it has difficulty to differentiate between one network which has almost zero value for part of the training data but large values for the rest, and another network with roughly the same \hat{F}(G) value but small values for all training data. The variance could help, but can not resolve this issue. This would be more important when the training data contains noise or outliers.
- It only concerns the generation of the training data, but not the sampled data from the network (at least not directly). Therefore it has no direct control of the fidelity of the generated samples.
- As shown by the authors, the proposed measure can be considered as the approximation of the true probability support not covered by the generative models, which also defines a necessary condition to avoid mode collapse. But what about the other part? It would have difficulty comparing two models with the same support but different high-density areas. Indeed, there are existing works which consider both the precision and recall of the generative models [1, 2, 3], and directly work with the generated samples instead of the training data. These should be discussed and compared with, not just the FID scores which have already been shown to have issues [3].

Some notations:
- In the last equation on Page 2,  should it be L_{G} instead of L_{D}?
- In the first equation on Page 3, should the denominator be N_{B} instead of N_{N}?
- "Optimality" in terms of generative models may depend on the downstream tasks. I do not think there exists a universal definition of "optimality" for generative models.

[1] M.S.M. Sajjadi, O. Bachem, M. Lucic, O. Bousquet, and S. Gelly. Assessing generative models via precision and recall. NeurIPS 2018.
[2] L. Simon, R. Webster, and J. Rabin. Revisiting precision and recall definition for generative model evaluation. ICML 2019.
[3] T. Kynkaanniemi, T. Karras, S. Laine, J. Lehtinen, and T. Aila. Improved precision and recall metric for assessing generative models. Arxiv:1904.06991.

**Experience Assessment:**

I have read many papers in this area.

**Review Assessment: Checking Correctness Of Derivations And Theory:**

N/A

**Review Assessment: Checking Correctness Of Experiments:**

I carefully checked the experiments.

**Review Assessment: Thoroughness In Paper Reading:**

I read the paper thoroughly.

---

> ### Author Response · Authors · 2019-11-13
> **Response to Reviewer #1**
>
> Thank you for the feedback on our paper. We address your concerns as follows:
>
> - It is true that if F>0 then we can still have memorization, because we are taking an average. In practice, this does not happen, as we captured in the histograms of Figure 4 and Figure 8 in the paper. We observe that the distribution of distance is relatively symmetric and unimodal, making the average a very informative measure of memorization. In addition, we are mostly concerned with having complete memorization of the data instead of just partial memorization. For partial memorization scenarios, we agree that variations on our metric (such as minimum distance) could be very useful as well.
>
> - We consider generating the training set as a first step toward understanding important issues like mode collapse. Of course, our measure alone will not be used to directly evaluate the fidelity of generated samples. High fidelity samples are desirable, but if F > 0 for these models, then that means that they are not learning the simplest distribution of all: the empirical distribution. Hence, generative models should have F = 0, which implies that there is no mode collapse with the empirical distribution.
>
> - We are comparing the support of G to the training set only (and not the probability densities), because we are focusing on the simpler, yet necessary, topic of memorization in generative networks. If a generative network cannot learn the training set, then there exists an image x such that the probability of generating x is equal to 0. Thus, a probabilistic distance of x from the distribution of Imag(G) is related to the distance between the image of G and x, which is our approach.
>
> Let us know if you have any further concerns about our paper, and thank you for the helpful feedback.

---

> > ### Comment · AnonReviewer1 · 2019-11-14
> > **Thanks for the response**
> >
> > Thanks for the detailed response.
> >
> > I think it is debatable to claim "generative models should have F = 0", especially when the generative models have covered the high-density areas.
> >
> > On the other hand, it is crucial to distinguish models which only memorize the training data but generate randomly beyond that and models which cover the data manifold well and generate reasonably. The proposed metric seems to favor the first than the latter, which is counter-intuitive. Therefore, metrics that cover both precision and recall would be more sensible in evaluation.

---

### Official Review · AnonReviewer3 · 2019-10-22
**Official Blind Review #3**

**Rating:** 3

**Review:**

This paper defines a goodness of fit measure F for generative networks, that reflects how well a model can generate the training data. F allows to detect mode collapse: as long as it is strictly positive, mode collapse is observed as parts of the training data have not been memorized. It aims at providing an alternative to the Fréchet Inception Distance and the Inception Score that rely on pretrained neural networks (whereas this new measure does not). It also provides insight into the DCGAN and WGAN networks in that regard, observing for instance that data subsampling helps decrease F, which motivates the use of a mixture of GANs.

This paper brings an interesting contribution to the evaluation of generative networks. However:

1.	The use of the square distance in the image space is not obvious and not justified.
2.	Computation of this metric is not straightforward: there is no theoretical guarantee and it is computationally expensive.
3.	The theoretical properties of this measure and its robustness are not investigated.
4.	Typos are obscuring the reading of the paper.

- Post rebuttal: I have read the authors' response and am maintaining my weak reject rating.

**Experience Assessment:**

I have published one or two papers in this area.

**Review Assessment: Checking Correctness Of Derivations And Theory:**

I assessed the sensibility of the derivations and theory.

**Review Assessment: Checking Correctness Of Experiments:**

I carefully checked the experiments.

**Review Assessment: Thoroughness In Paper Reading:**

I read the paper at least twice and used my best judgement in assessing the paper.

---

> ### Author Response · Authors · 2019-11-13
> **Response to Reviewer #3**
>
> Thank you for the feedback on our paper. We address your concerns as follows:
>
> 1. Defining a metric between images is a long-standing problem in image processing, because standard p-norms do not capture image structure well. We use squared distance (2-norm) because it has a long history in not only signal and image processing but beyond. Nevertheless, other metrics can be used within our framework if there is prior information that implies that a particular metric would be superior to others. For this reason, we do not claim that the 2-norm is optimal for images.
>
> 2. We follow standard methods for optimizing over the latent space of a generative network such those from the GLO paper. This enables us to have a certain confidence in the optimization problem because this method “... recovers the true latent vector 100% of the time to arbitrary precision.” [1] Although, this is not theoretical guarantees, their empirical performance is a convincing argument for why they should be used for calculating F in practice. Developing theoretical guarantees for this nonconvex optimization problem is beyond the scope of this paper and would be an interesting paper by itself.
>
> 3. The robustness of calculating F is measured with different initializations as shown in Figure 6. If we optimize over the latent space with different initial distributions of z, we find that the statistics of the solution z* are the same. Hence, the calculation of F is robust to different initial distributions of z, which means that even unlikely z’s will converge to a z* that has typical statistics.
>
> Let us know if you have any further concerns about our paper, and thank you for the helpful feedback.
>
> [1] Zachary C Lipton and Subarna Tripathi.  Precise recovery of latent vectors from generative adversarial networks. arXiv preprint arXiv:1702.04782, 2017.

---

### Decision · Program_Chairs · 2019-12-19

**Decision:**

Reject

**Comment:**

This paper proposes to measure the distance of the generator manifold to the training data. The proposed approach bears significant similarity to past studies that also sought to analyze the behavior of generative models that define a low-dimensional manifold (e.g. Webster 2019, and in particular, Xiang 2017). I recommend that the authors perform a broader literature search to better contextualize the claims and experiments put forth in the paper.

The proposed method also suffers from some limitations that are not made clear in the paper. First, the measure depends only on the support of the generator, but not the density. For models that have support everywhere (exact likelihood models tend to have this property by construction), the measure is no longer meaningful. Even for VAEs, the measure is only easily applicable if the decoder is non-autoregressive so that the procedure can be applied only to the mean decoding.

In this current state, I do not recommend the paper for submission.

Xiang (2017). On the Effects of Batch and Weight Normalization in Generative Adversarial Networks
Webster (2019). Detecting Overfitting of Deep Generative Networks via Latent Recovery